# The ART-Rsp5 ubiquitin ligase network comprises a plasma membrane quality control system that protects yeast cells from proteotoxic stress

Yingying Zhao[†], Jason A MacGurn[†], Max Liu, Scott Emr*

Weill Institute of Cell and Molecular Biology, Cornell University, Ithaca, United States

**Abstract** Secretory cargo that cannot fold properly in the ER are selectively targeted for removal by a well-studied ER-associated degradation pathway, or ERAD. In contrast, very little is known about post-ER quality control mechanisms for damaged or misfolded integral membrane proteins. Here we describe a quality control function of the Rsp5-ART ubiquitin ligase adaptor network that functions to protect plasma membrane (PM) integrity. Failure to mediate this protective response during heat stress leads to toxic accumulation of misfolded integral membrane proteins at the cell surface, which causes loss of PM integrity and cell death. Thus, the Rsp5-ART network comprises a PM quality control system that works together with sequential quality control pathways in the ER and Golgi to (i) target the degradation of proteins that have exceeded their functional lifetime due to damage and/or misfolding and (ii) limit the toxic accumulation of specific proteins at the cell surface during proteotoxic stress.

*For correspondence: sde26@cornell.edu

†These authors contributed equally to this work

Competing interests: The authors declare that no competing interests exist.

## Introduction

Cells have evolved elaborate protein quality control mechanisms that assist proteins in achieving their desired native structure and also help target the degradation of proteins that have become misfolded, damaged, or otherwise aberrant. This system, often referred to as the cellular proteostasis network, consists of an elaborate network of proteins dedicated to preventing the toxic accumulation of unfolded or misfolded proteins. Newly translated proteins interact with chaperones that direct their folding into the native active form. Proteins that do not fold correctly or older proteins that exceed their functional lifetime are recognized by cellular quality control mechanisms that target them for degradation. Often, the cell employs compartment-specific mechanisms that ensure protein quality control. For example, misfolded protein domains that are soluble and accessible to the cytosol can be targeted for degradation by CHIP, an E3 ligase that interacts with various chaperones to target ubiquitination of misfolded substrates (*Meacham et al., 2001*). Similarly, unfolded soluble protein domains in the lumen of the ER are clients of the Hsp70 family member BiP/Kar2, and extended association with BiP/Kar2 can lead to targeting for ubiquitination by the ERAD machinery and subsequent proteasomal degradation (*Claessen et al., 2011*; *Walter and Ron, 2011*). In these examples, targeting of misfolded proteins for ubiquitination and subsequent degradation is mediated by prolonged interaction with chaperones, which serve as adaptors that recruit E3 ubiquitin ligases to modify misfolded clients. In both cases, ubiquitinated substrates are ultimately degraded by the proteasome.

Quality control of integral plasma membrane proteins plays a critical role in many human diseases because it affects the quantity of functional proteins—ion channels, nutrient transporters, and signaling receptors—at the cell surface. In contrast to proteins in the cytosol or the ER, which are degraded by the proteasome, most post-ER integral membrane proteins are degraded via lysosomal trafficking. This process is typically ubiquitin-dependent and involves: (i) selective recognition of cargo for

**eLife digest** Cells have evolved elaborate mechanisms for the detection of misfolded or damaged proteins, and for targeting their degradation. Since the accumulation of misfolded proteins is toxic to the cell, these protein quality control systems are critical for the maintenance of normal cellular function over the lifetime of an organism. The breakdown of this quality control correlates with the progression of neurodegenerative disorders including Alzheimer's, Huntington's and Parkinson's disease. Normal function of the protein quality control machinery can also cause disease: this is the case with channelopathies such as cystic fibrosis, in which mutant ion channels are targeted for degradation and therefore cannot function correctly at the cell surface. Understanding how protein quality control systems recognize misfolded proteins and target their degradation, and designing ways to stabilize or destabilize specific targets, particularly at the cell surface, could thus lead to the development of new therapeutic strategies.

While protein quality control mechanisms in the cytosol and endoplasmic reticulum (ER) have been studied extensively, much less is known about quality control of integral membrane proteins after they exit the ER. Maintaining the quality of cell surface proteins impacts many critical biological functions including nutrient uptake, signaling and the functioning of specialized surface structures such as cell junctions.

Here, Zhao et al. describe a new quality control mechanism that prevents misfolded proteins from accumulating in the plasma membrane. Building upon earlier work describing a network of adaptor proteins (called ARTs) for the Rsp5 ubiquitin ligase, Zhao et al. show that subjecting cells to proteotoxic stress, particularly thermal stress, triggers ART-Rsp5-mediated clearance of misfolded plasma membrane proteins. When ART-Rsp5-mediated clearance is abrogated, misfolded proteins accumulate at the cell surface, resulting in a rapid loss of cellular integrity. In the brain, such proteotoxicity can lead to cell death and neurodegeneration, thereby highlighting the importance of this plasma membrane quality control system.

ubiquitination, (ii) capture of ubiquitinated cargo for trafficking to the endosome, (iii) endosomal sorting by ESCRT complexes, which package ubiquitinated cargo into vesicles that bud into the lumen of the endosome (intralumenal vesicles, or ILVs), and (iv) fusion of these multivesicular endosomes with lysosomal/vacuolar compartments, resulting in cargo degradation (*Henne et al., 2011*; *MacGurn et al., 2012*). While the sorting and trafficking mechanisms in this elaborate downregulation process have been studied extensively, the initial step of cargo recognition for ubiquitination, particularly with respect to the recognition of misfolded proteins in post-ER compartments, is still poorly understood.

Recently, a peripheral membrane protein quality control system was identified for the detection, ubiquitination, and downregulation of misfolded CFTR from the plasma membrane. This mechanism involves recognition of the misfolded cytosolic domain of CFTR by molecular chaperones (Hsc70 and Hsp90) which then recruit the E3 ubiquitin ligase CHIP to ubiquitinate misfolded CFTR (*Okiyoneda et al., 2010*). Importantly, a similar mechanism was shown to function in the endocytic downregulation of artificial PM cargo fused to cytosolic domains that undergo temperature-induced misfolding (*Apaja et al., 2010*). These studies provide an elegant mechanism for recognition and clearance of membrane proteins with extensive soluble cytosolic domains but do not address how aberrant conformations of membrane spanning domains are detected and degraded. Evidence for other quality control mechanisms at the PM has been reported using many different PM cargoes. For example, early studies of Pma1, the yeast plasma membrane H+-ATPase, report its rapid degradation following heat stress (*Piper, 1995*). More recent studies have proposed that an unstable mutant allele of Pma1 which is rapidly ubiquitinated, endocytosed, and degraded in the yeast vacuole may represent a misfolded form of the protein (*Liu and Chang, 2006*). Other studies of the yeast general amino acid transporter Gap1 demonstrate that loss of sphingolipids alters the activity and stability of Gap1 at the cell surface, suggesting that proteins may become misfolded when PM lipid composition is altered (*Lauwers et al., 2007*). However, mechanisms for the recognition and targeted ubiquitination of misfolded integral membrane proteins have remained elusive. Furthermore, the physiological consequences of the accumulation of misfolded proteins at the cell surface remain unclear.

Previously, we identified and characterized a network of Rsp5 ubiquitin ligase adaptors called ARTS, or arrestin-related trafficking adaptors. The ART modular adaptor network mediates the recognition and ubiquitination of cell surface proteins, thereby directing the endocytic remodeling of PM protein composition. Thus, ART proteins serve as a key point of regulation during changes in nutrient availability and environmental stress. Here, we report that various proteotoxic stresses trigger extensive cell surface remodeling characterized by the endocytic downregulation of various different PM cargo proteins. Specifically, we show that heat stress results in misfolding of certain thermolabile PM cargo molecules which are targeted for endocytosis and vacuolar degradation by the Rsp5-ART adaptor network. Defects in this endocytic targeting system result in accumulation of cell surface proteins that is toxic during heat stress. This toxicity is associated with loss of PM integrity, which we propose is caused by conformational flux or misfolding of integral membrane proteins at the cell surface. The ART-Rsp5 quality surveillance mechanism at the PM functions in parallel with both ER and Golgi QC systems to control both the quality and quantity of integral membrane proteins in the cell.

## Results

### Heat stress triggers extensive endocytic downregulation

Previously, we found that certain environmental changes elicit highly-specific surface remodeling programs, such as the methionine-induced endocytic down-regulation of the methionine transporter Mup1 (*Lin et al., 2008*). In contrast, other environmental changes can stimulate global surface remodeling programs, such as the TORC1-mediated tuning of PM protein composition by activation of the ART adaptor network (*MacGurn et al., 2011*). Given the well-characterized effect of high temperature on protein denaturation, we reasoned that the conformational flux facilitated by heat stress might trigger partial denaturation or misfolding of integral membrane proteins, providing a simple assay for the investigation of PM quality control mechanisms. To characterize the effect of heat stress on integral PM protein stability in yeast, we analyzed the trafficking and degradation of various PM cargoes in cells grown at 26°C and shifted to 38°C for 60 min. In general, most cargoes were observed to undergo endocytosis followed by vacuolar trafficking (*Figure 1A* and *Figure 1—figure supplements 1 and 2*) and degradation (*Figure 1B* and *Figure 1—figure supplements 3–5*), although a few cargoes appeared to be stable (Pdr5, *Figure 1B* and *Figure 1—figure supplement 5*) or induced during heat stress (*Figure 1—figure supplement 6*). This heat-induced endocytic downregulation was observed with varying kinetics and efficiency for many diverse cargoes including amino acid transporters (Mup1, Lyp1, Dip5), hexose transporters (Hxt3), G-protein coupled receptors (Ste3) and proton pumps (Pma1) (*Figure 1B*). Similar to heat stress, we found that other known proteotoxic stresses (growth in 10% ethanol, 5 mM DTT, or 2.5 mM diamide) triggered the endocytosis and vacuolar trafficking of different PM cargoes, suggesting that endocytic clearance may be associated with protein damage or misfolding. For example, ethanol stress triggered the endocytic downregulation of the arginine transporter Can1 and the methionine transporter Mup1, but had little effect on the surface stability of Lyp1 (*Figure 1—figure supplement 7*). In contrast, oxidative stress triggered the endocytic downregulation of Lyp1 and Can1 but had little effect on the surface stability of Mup1 (*Figure 1—figure supplement 7*). Given its dramatic affect on surface protein stability, and given the ability of *Saccharomyces cerevisiae* to tolerate a wide range of temperatures for growth (*Steinmetz et al., 2002*), we decided to further investigate the molecular basis for heat-induced endocytic downregulation.

To further characterize the thermostability of specific cargoes, we grew yeast cells at 26°C and measured the kinetics of cargo degradation following shift to different temperatures (26°C, 34°C, 38°C, 40°C, 42°C) all within the normal growth range for *S. cerevisiae*. We found that different cargoes exhibited a range of thermostabilities (*Table 1* and *Figure 1—figure supplement 8*) suggesting heat stress triggers thermo-instability that is cargo-intrinsic. Since conformational flux and protein misfolding both increase as a function of temperature, we reasoned that heat-induced instability of PM proteins may relate to misfolding or conformational instability caused by increased temperature. To test this idea, we performed thermostability analysis of the highly thermo-labile lysine transporter Lyp1 in the presence of glycerol, which is known to promote/stabilize protein folding and function as a chemical chaperone (*Bernier et al., 2004*). Importantly, glycerol enhanced the thermostability of Lyp1 (*Figure 1C* and *Figure 1—figure supplements 4 and 9*) but did not stabilize the methionine-induced endocytic downregulation of the methionine transporter Mup1 (*Figure 1—figure supplement 10*). These data demonstrate that glycerol can stabilize certain integral membrane proteins during heat

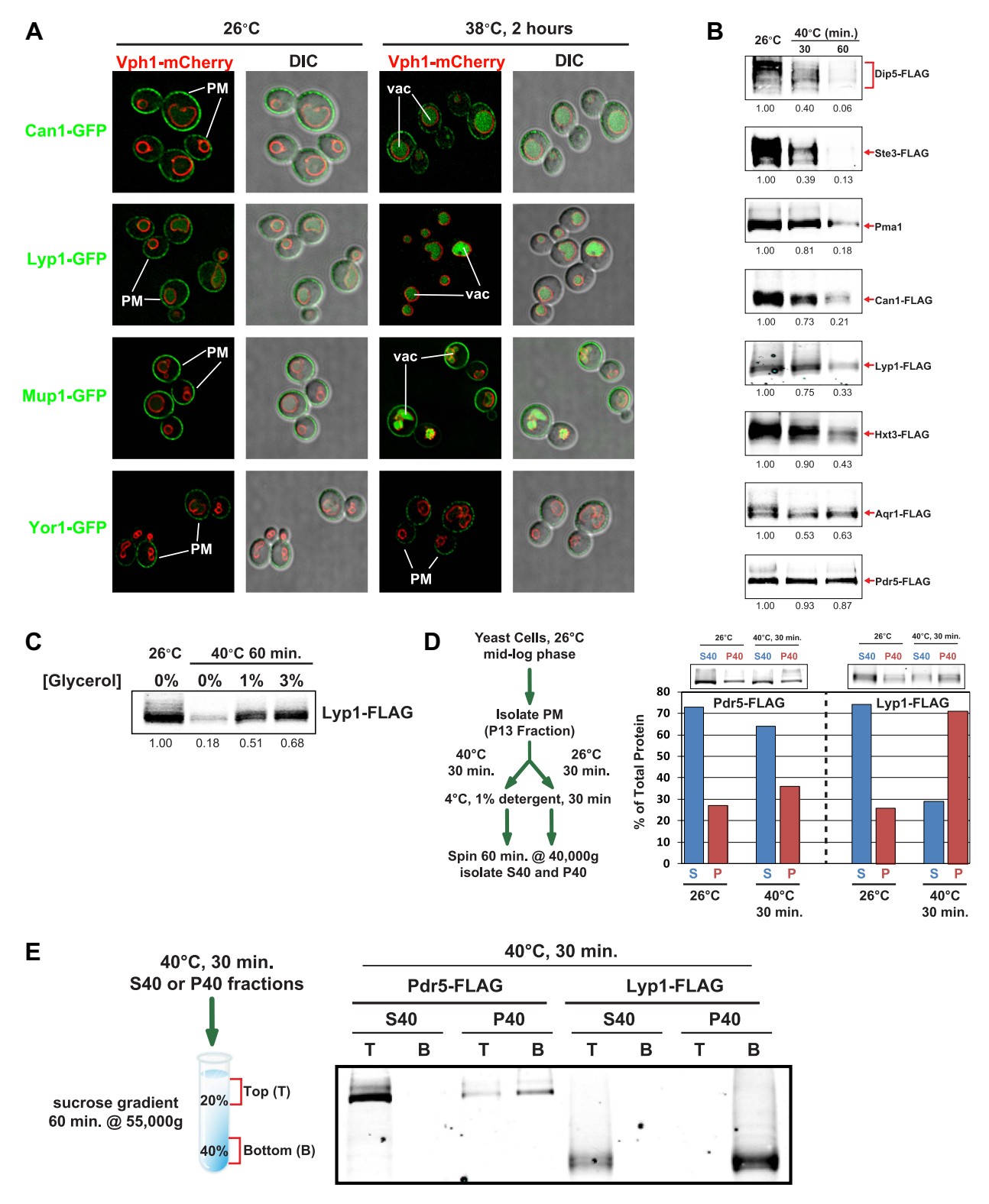

**Figure 1**. Heat stress triggers endocytic downregulation. (**A**) Fluorescence distribution of GFP-tagged endocytic cargoes (green) was analyzed in wildtype yeast cells expressing the vacuolar marker Vph1-mCherry (red). Cells were grown to mid-log phase at 26°C (left panels) and then shifted to 38°C for 2 hr (right panels). Plasma membrane ('PM') and vacuole ('vac') localization are indicated. Note that the GFP moiety of cargo fusions resists hydrolysis and thus fluorescence signal accumulates in the vacuole even as the protein appears to be degraded as monitored by immunoblot. (**B**) Stability of affinity-tagged cargoes was analyzed following temperature shift from 26°C (left lane) to 40°C. The number beneath each lane indicates
*Figure 1. Continued on next page*

*Figure 1. Continued*

quantification of protein abundance (relative to 26°C, t = 0 and normalized to G6PDH as a loading control) determined using the Li-Cor system. (**C**) Analysis of heat-induced degradation of Lyp1 in the presence of glycerol, a chemical chaperone. (**D**) Detergent solubility of a thermostable cargo (Pdr5) and a thermolabile cargo (Lyp1) was analyzed at low and high temperature. A schematic representation for the experimental design is shown at the left. (**E**) Detergent soluble (S40) and insoluble (P40) fractions from membranes incubated at 40°C for 30 min were analyzed for mobility on sucrose step gradients. Top (T) fractions were immobile on the gradient, while bottom (B) fractions migrated through the gradient.

The following figure supplements are available for figure 1:

**Figure supplement 1**. Cell surface fluorescence intensity was measured for Mup1-GFP and Can1-GFP following growth at 26°C or following a shift to 38°C for 2 hr.

**Figure supplement 2**. Mup1-pHluorin was used to quantify surface abundance of Mup1 for a population of yeast cells at 26°C or following a shift to 38°C for 2 hr (bottom panel).

**Figure supplement 3**. Stability of affinity-tagged Mup1 and Pma1 was analyzed following temperature shift from 26°C to indicated temperatures.

**Figure supplement 4**. Stability of Lyp1 was analyzed following temperature shift from 26°C to indicated temperatures in the presence or absence of 3% glycerol.

**Figure supplement 5**. Stability of affinity-tagged Aqr1 and Pdr5 was analyzed following temperature shift from 26°C to indicated temperatures.

**Figure supplement 6**. Identification of integral PM proteins induced by shifting cells to 40°C.

**Figure supplement 7**. Various proteotoxic stresses trigger endocytic downregulation.

**Figure supplement 8**. Analysis of cargo thermostability.

**Figure supplement 9**. Analysis of Lyp1 thermostability in the absence or presence of 3% glycerol.

**Figure supplement 10**. Analysis of the kinetics of Mup1 lysosomal degradation following the addition of methionine at 26°C in the absence or presence of 3% glycerol.

stress but has no effect on substrate-induced endocytosis. Together, these data are consistent with a mechanism whereby heat stress induces different degrees of misfolding in distinct PM proteins and this conformational instability of integral PM proteins results in their selective recognition and subsequent targeting for endocytic downregulation.

We decided to further explore the possibility that the thermoinstability observed for certain cargo at the PM is a direct result of misfolding during heat stress. First, we tested for changes in the physico-chemical properties of a relatively stable cargo (Pdr5) and a comparatively heat-labile cargo (Lyp1) using a detergent solubilization assay. Specifically, we isolated yeast plasma membranes (P13 fraction), incubated them at 26°C or 40°C, and analyzed the percentage of cargo that is detergent soluble. While Pdr5 did not exhibit a significant change in detergent solubility at high temperature (*Figure 1D*, left), Lyp1 detergent solubility was significantly reduced during heat stress (*Figure 1D*, right) indicating that Lyp1, but not Pdr5, may be misfolding or aggregating at high temperature. To further explore this possibility, we analyzed S40 (detergent soluble) and P40 (detergent insoluble) fractions from membranes subject to heat stress (40°C for 30 min) by scoring mobility on a sucrose gradient (*Figure 1E*). Importantly, while detergent-soluble cargo in the S40 floated on the sucrose gradient, detergent-insoluble Lyp1 in the P40 fraction migrated to the bottom of the sucrose gradient (*Figure 1E*), suggesting Lyp1 aggregation during heat stress. These data demonstrate that a thermolabile cargo (Lyp1) tends to aggregate at high temperature and that the rapid degradation of such cargo in response to heat stress may protect cells from the accumulation of aggregated proteins at the surface.

## The heat-induced endocytic response is mediated by the Rsp5 ubiquitin ligase

To identify the molecular mechanism that targets misfolded cargo for endocytosis during heat stress, we first tested if this stress response requires Rsp5, a yeast Nedd4 family ubiquitin ligase that targets

**Table 1.** Cargo thermostability

| Cargo | 34°C | 38°C | 40°C | 42°C |
|---|---|---|---|---|
| Lyp1 | 160 | 50 | 25 | – |
| Pma1 | Stable | 95 | 45 | – |
| Mup1 | Stable | 100 | 65 | – |
| Aqr1 | – | Stable | 100 | 70 |
| Pdr5 | – | Stable | 400 | 200 |

Kinetic analysis of heat-induced cargo degradation (*Figure 1—figure supplements 3 and 4*) was used to estimate half-lives of each cargo at each temperature tested. Half-lives are indicated in minutes. '–' indicates half-life not determined.

many PM cargo for ubiquitination and endocytosis (*MacGurn et al., 2012*). To do this, we analyzed heat-induced cargo trafficking in yeast strains expressing various mutant alleles of Rsp5. Like all Nedd4 family members, Rsp5 contains a C-terminal HECT ubiquitin ligase domain and multiple WW domains required for substrate targeting. These WW domains direct substrate targeting for ubiquitination by binding to a specific peptide sequence called a PY motif (PPXY) present in adaptor proteins (ARTs, ARRDCs) and some substrates (ENaC). While the WW1 domain of Rsp5 was dispensable for heat-induced cargo degradation, mutations in either WW2 or WW3 significantly abrogated the endocytic response (*Figure 2A–C*). Since these mutations result in significant accumulation of cargo at the PM during heat stress, we tested if mutant alleles of *RSP5* exhibited defects in thermotolerance by scoring for growth at elevated temperatures. While *rsp5-ww1* mutants did not exhibit any observable temperature sensitivity defect, *rsp5-ww2* and *rsp5-ww3* mutants were extremely temperature sensitive and failed to grow at 38°C (*Figure 2D*).

To better understand the basis of this temperature sensitive phenotype, we utilized a polar fluorescent vital dye, propidium iodide (PI, MW = 668.4 Da), which binds to nucleic acids but is membrane impermeant. Thus, PI staining can be used to score the number of cells in a population that have lost cellular integrity. For example, wildtype yeast cells grown at 26°C exhibit negligible (<1%) PI-positive cells within the population (*Figure 3A*). However, if these same yeast cells are heat-shocked (65°C for 10 min) or treated with the drug nystatin, which binds to ergosterol and forms pores in the yeast plasma membrane, all yeast cells (100%) stain positive for PI (*Figure 3A*), demonstrating the utility of PI as a marker for PM integrity. Next, we used flow cytometry to analyze PI staining in wildtype and mutant cells following heat stress. After three hours at 40°C, wildtype and *rsp5-ww1* yeast cells did not exhibit any significant PI staining (*Figure 3B,C*). In contrast, *rsp5-ww2* and *rsp5-ww3* mutant cells exhibited a significant fraction of the population that stained PI-positive in response to heat stress (*Figure 3B,C*). These results suggest that the rapid loss of PM integrity of *rsp5-ww2* and *rsp5-ww3* mutant cells during heat stress is either the basis of thermosensitivity or a secondary effect resulting from cell death.

Heat stress broadly induces protein misfolding and denaturation across all cellular compartments. We next wanted to determine if the observed loss of PM integrity in *rsp5* mutants during heat stress is linked to the accumulation of misfolded cargo specifically at the cell surface. To test this, we analyzed PI staining following heat stress for mutants that affect different stages of the endocytic pathway (*Figure 3C*). While mutations that abrogate endocytosis (*Δend3*, *Δrvs167*) resulted in significant PI staining in response to heat stress, mutant cells defective for downstream trafficking events such as endosomal sorting and multivesicular body biogenesis (*Δvps23*), endosomal-vacuolar fusion (*Δvam3*), and vacuolar degradation (*Δpep4*) exhibited negligible PI staining in response to heat stress (*Figure 3C*). These results indicate that plasma membrane clearance is the critical event protecting cells from heat-induced loss of PM integrity, while downstream sorting and trafficking events are dispensable. Indeed, the misfolding of integral PM proteins, many of which are channels and transporters designed to facilitate import of nutrients into the cell, could contribute to loss of critical ion gradients if not targeted for rapid clearance from the PM. Thus, PM quality control is an important line of defense for maintaining the integrity of the cell.

## The ART adaptor network protects PM integrity during heat stress

In many cases, Rsp5 ubiquitin ligase activity is directed by adaptor proteins which target specific cargoes at the PM in response to various stimuli. To test if any Rsp5 adaptor proteins function in targeting cargoes for endocytic downregulation during heat stress, we screened a set of yeast knockout strains to identify Rsp5 adaptors required for thermotolerance. While most ART (*arrestin-related trafficking adaptor*) family proteins were found to be dispensable for growth at high temperatures, *Δart1* yeast cells were temperature sensitive for growth at 38°C similar to *rsp5-ww2* and *rsp5-ww3* mutant cells

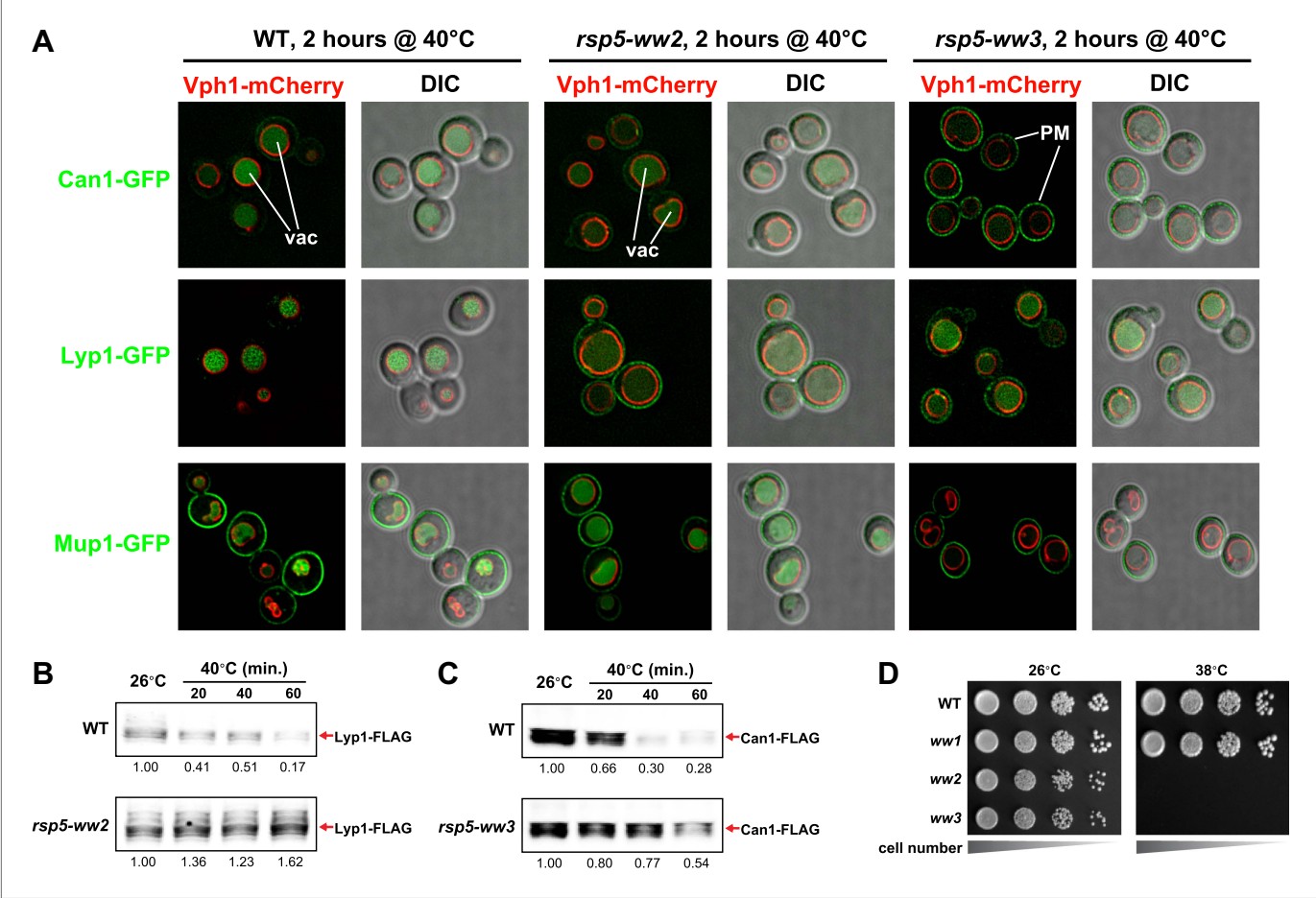

**Figure 2**. Rsp5 mediates the heat-induced endocytic response. (**A**) Fluorescence distribution of GFP-tagged endocytic cargoes (green) was analyzed in wildtype (left panels), *rsp5-ww2* (middle panels), or *rsp5-ww3* (right panels) yeast cells expressing the vacuolar marker Vph1-mCherry (red). Cells were grown to mid-log at 26°C and then shifted to 38°C for 2 hr. Plasma membrane ('PM') and vacuole ('vac') localization are indicated. (**B**) and (**C**) Stability of affinity-tagged cargoes was analyzed following temperature shift from 26°C (left lane) to 40°C. The number beneath each lane indicates quantification of protein abundance (relative to 26°C, t = 0 and normalized to G6PDH as a loading control) determined using the Li-Cor system. (**D**) Heat-sensitivity analysis of wildtype, *rsp5-ww2*, or *rsp5-ww3* yeast cells.

(*Figure 4A* and *Figure 4—figure supplement 1*). Importantly, our analysis revealed that heat-induced endocytosis and vacuolar degradation of most cargoes is Art1-independent, with the exception of the lysine transporter Lyp1 (*Figure 4B*). In contrast to other cargoes, Lyp1 endocytosis and vacuolar degradation is abrogated in the absence of Art1 (*Figure 4B,C*), underscoring the cargo-specific role Art1 plays in the heat-induced endocytic response. Consistent with its role in the heat-induced endocytic response, we found that Art1-GFP translocates to the PM in response to heat stress (*Figure 4—figure supplement 2*). Thus, while *rsp5* mutant strains exhibit broad defects in the heat-induced endocytic response (*Figure 2*), Δ*art1* mutant cells are defective for the turnover of only a subset of PM proteins.

Our results indicate that the temperature sensitive growth phenotype exhibited by *rsp5-ww2*, *rsp5-ww3*, and Δ*art1* mutant cells likely results from accumulation of misfolded proteins in the plasma membrane, which ultimately triggers loss of PM integrity and cell death. Based on this rationale, we performed a screen to identify bypass suppressors of the Δ*art1* mutant temperature sensitive phenotype in order to uncover other components of the ART-Rsp5 network or new pathways for PM protein quality control. All multicopy overexpression plasmids that suppressed the Δ*art1* temperature sensitive growth defect (84 total suppressors isolated) were found to encode either *ART1* (35 individual isolates) or *ART2* (49 individual isolates) (*Figure 4—figure supplement 3*). The identification of *ART2* as a suppressor of the Δ*art1* temperature sensitivity phenotype indicated that (i) multiple ART family

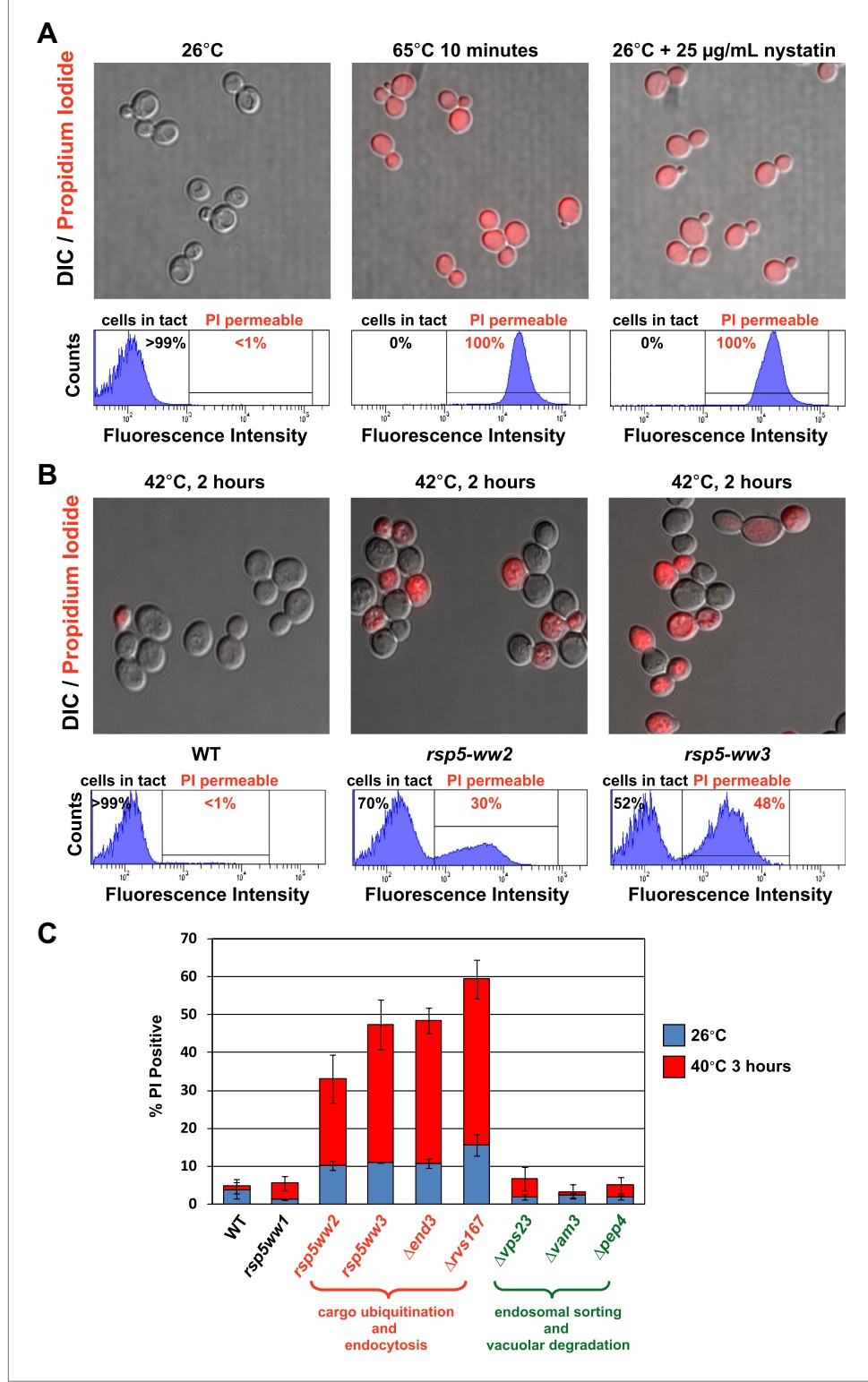

**Figure 3**. Rsp5-mediated endocytic clearance protects PM integrity during heat stress. (**A**) Yeast cells that were either grown at 26°C to mid-log phase (left), heated to 65°C for 10 min (middle), or treated with nystatin (right) were stained with propidium iodide (PI) and analyzed by fluorescence microscopy (top) and flow cytometry (bottom). (**B**) Wildtype (left), *rsp5-ww2* (middle), or *rsp5-ww3* (right) yeast cells were grown to mid-log at 26°C, shifted to 42°C for 2 hr, stained with PI, and analyzed by fluorescence microscopy (top) and flow cytometry (bottom). (**C**) Flow cytometry was used to analyze PI staining of the indicated strains at 26°C (blue bars) or following growth at 40°C for 3 hr (red bars).

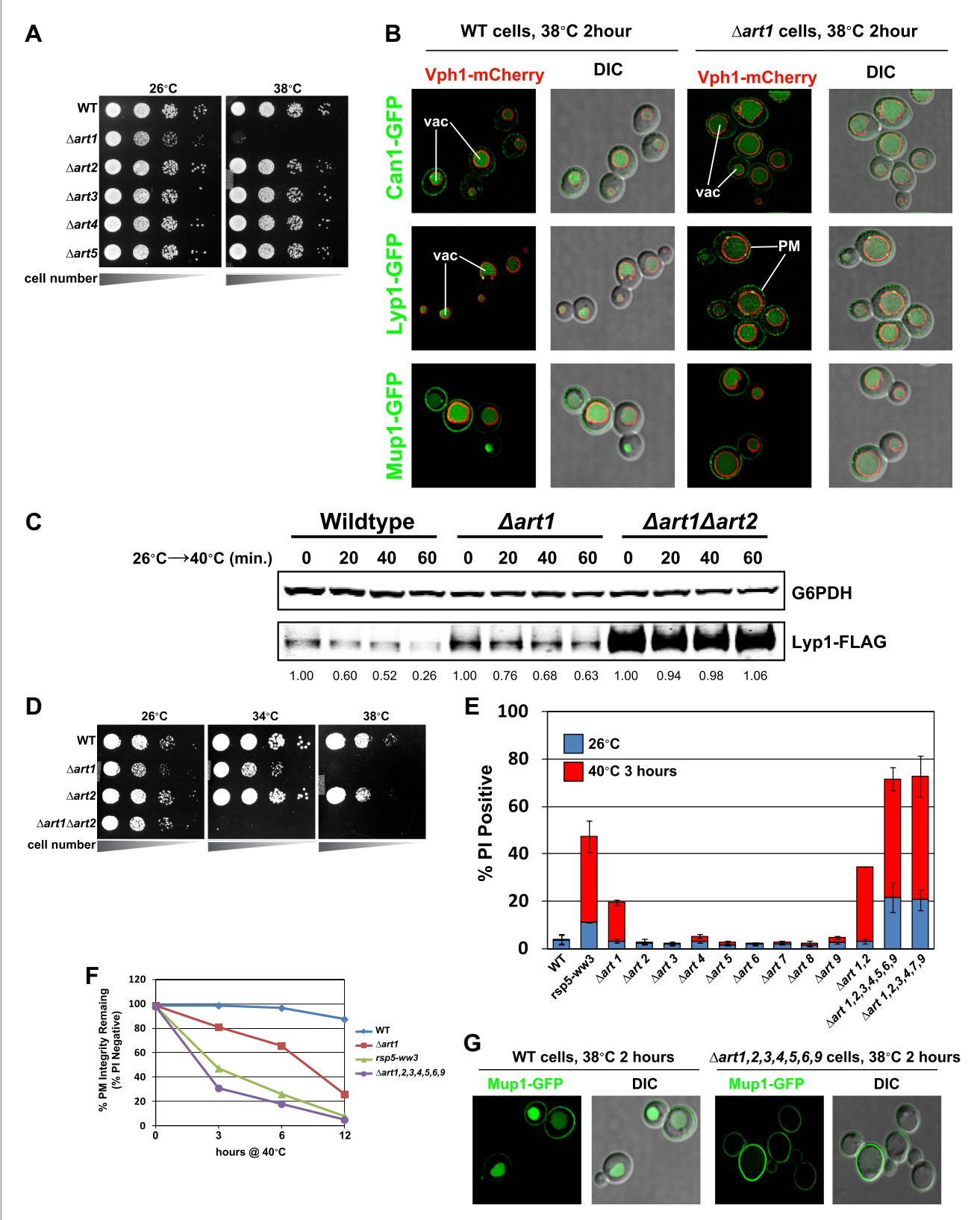

**Figure 4**. ARTs protect plasma membrane integrity during heat stress. (**A**) Heat-sensitivity analysis of wildtype and *art* mutant yeast cells. See *Figure 4—figure supplement 1* for additional characterization of *art* mutant yeast cells. (**B**) Fluorescence distribution of GFP-tagged endocytic cargoes (green) was analyzed in wildtype (left panels) and Δ*art1* (right panels) yeast cells expressing the vacuolar marker Vph1-mCherry (red). Cells were grown to

*Figure 4. Continued on next page*

*Figure 4. Continued*
mid-log at 26°C and then shifted to 38°C for 2 hr. Plasma membrane ('PM') and vacuole ('vac') localization are indicated. (**C**) Stability of affinity-tagged Lyp1 was analyzed following temperature shift from 26°C (left lane) to 40°C. The number beneath each lane indicates quantification of protein abundance (relative to 26°C, t = 0) determined using the Li-Cor system. (**D**) Heat-sensitivity analysis of wildtype, *Δart1*, *Δart2*, and *Δart1Δart2* mutant yeast cells. (**E**) Flow cytometry was used to analyze PI staining of the indicated strains at 26°C (blue bars) or following growth at 40°C for 3 hr (red bars). (**F**) Kinetic analysis of PI staining for a 40°C heat stress timecourse. (**G**) Fluorescence distribution of GFP-tagged Mup1 (green) was analyzed in wildtype (left panels) and *Δart1Δart2Δart3Δart4Δart5Δart6Δart9* (right panels) yeast cells during heat stress.
The following figure supplements are available for figure 4:
**Figure supplement 1**. Heat-sensitivity analysis of wildtype and *art* mutant yeast cells.
**Figure supplement 2**. Fluorescence distribution of GFP-tagged Art1 (green) was analyzed at 26°C and 38°C in yeast cells expressing the vacuolar marker Vph1-mCherry (red).
**Figure supplement 3**. Results of a screen to identify multicopy bypass suppressors of the *Δart1* ts phenotype.

proteins may protect cells during heat stress and (ii) different ART proteins may have overlapping cargo specificities which create redundancy in the system. Although *Δart2* mutant cells did not exhibit any temperature sensitive phenotypes, we found that deletion of *ART2* in *Δart1* mutant cells resulted in an enhanced temperature sensitivity phenotype (*Figure 4D*) that correlated with increased Lyp1 accumulation in the *Δart1Δart2* double mutant (*Figure 4C*).

To quantitatively measure the protective role of the ART adaptor network, we combined flow cytometry analysis and the PI staining assay to score a panel of *ART* deletion mutants for loss of PM integrity during heat stress. By limiting the time of exposure to heat stress (40°C for only 3 hr) we aimed to detect the onset of PM permeability defects associated with accumulation of cargo at the surface. Although *Δart1* mutant cells exhibited a slight loss of PM integrity (~18% of the cells are PI-positive following 40°C for 3 hr), the other adaptor mutants tested exhibited no defects in PM integrity in response to heat stress (*Figure 4E*). However, striking synthetic defects were observed when *Δart1* was combined with deletion of other *ART* genes (*Figure 4E,F*). Furthermore, while ART1 and ART2 are dispensable for Mup1 trafficking to the vacuole in response to heat stress (*Figure 4B* and data not shown), the loss of multiple ART genes effectively blocks Mup1 turnover in response to heat stress (*Figure 4G*). These results underscore the critical role of the ART adaptor network in protecting PM integrity during heat stress.

## Cell surface protein accumulation is toxic during heat stress

Our observations that Rsp5 and the ART adaptor network are critical mediators of the heat-induced endocytic response suggested that these proteins may function as a critical quality control mechanism at the PM. Given that loss of this response during heat stress results in (i) cargo accumulation at the PM, (ii) loss of PM integrity as defined by the permeabilization of the cell to a large (668.4 Da) polar dye (PI) and (iii) loss of cell viability, we hypothesized that heat stress triggers conformational instability or misfolding of integral membrane proteins at the cell surface and that failure to recognize, remove and degrade these proteins threatens the integrity of the plasma membrane. Based on our observation that *Δart1* mutant cells, which are temperature sensitive, exhibit specific accumulation of the lysine transporter Lyp1 during heat stress, we decided to test if Lyp1 accumulation is toxic at high temperature. To do this, we analyzed temperature sensitivity of *Δart1Δlyp1* mutant cells and, to our surprise, found that loss of Lyp1 partially suppressed the temperature sensitive phenotype of *Δart1* mutant cells (*Figure 5A*). This result suggests that Lyp1 accumulation at the PM during heat stress is toxic and significantly contributes to the temperature sensitive phenotype of *Δart1* mutant cells.

To explore this idea, we titrated Lyp1 expression using several different promoters to determine if Lyp1 accumulation in *Δart1Δlyp1* cells is toxic during heat stress. Importantly, we found that Lyp1 expression correlated with temperature sensitivity in *Δart1Δlyp1* cells, where the expressed Lyp1 could not be efficiently internalized in response to heat stress (*Figure 5B*). This same Lyp1 titration failed to cause temperature sensitivity in wildtype cells where Lyp1 undergoes heat-induced endocytosis, but higher levels of Lyp1 expression using the TDH3 promoter did confer temperature sensitivity

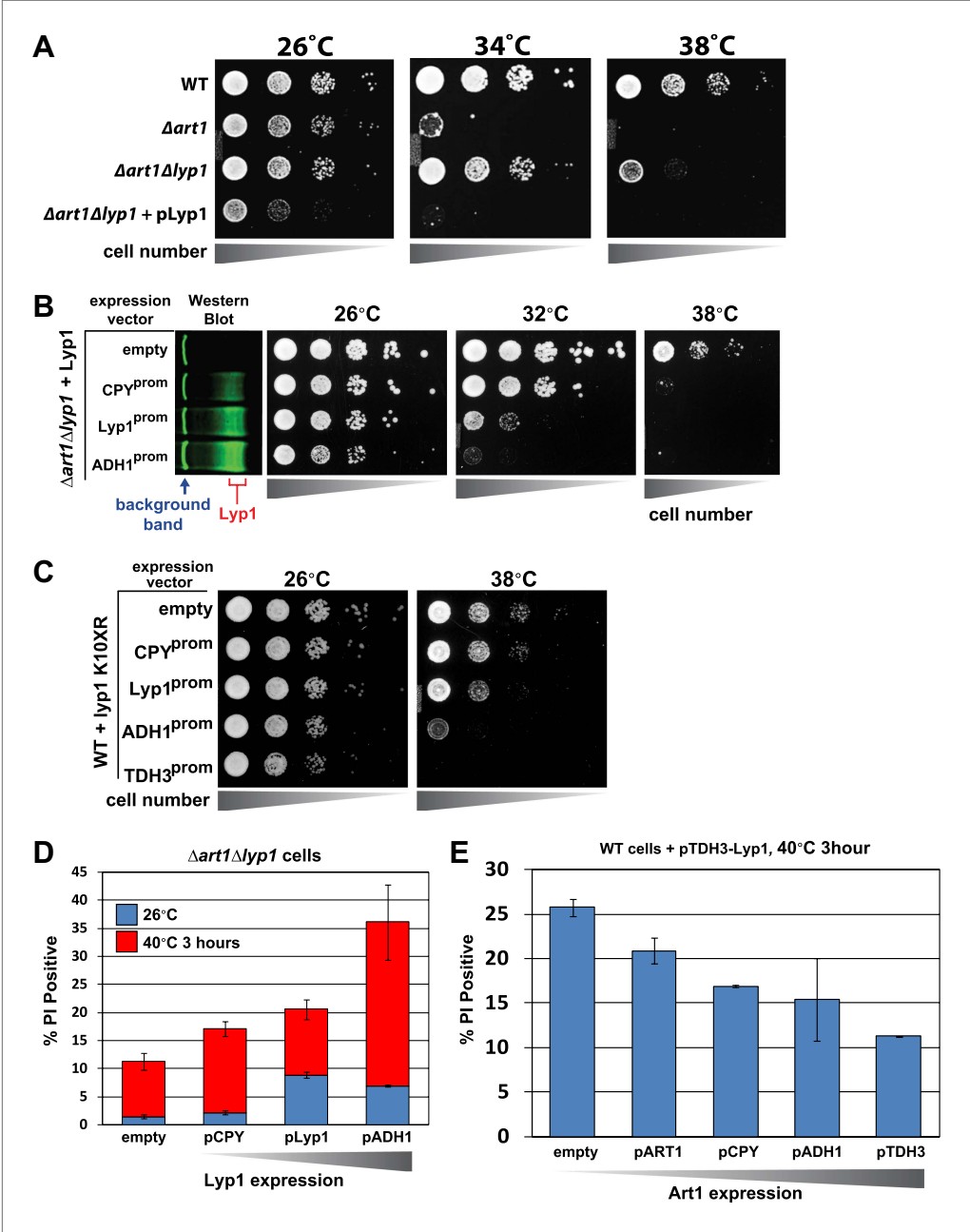

**Figure 5**. Cargo accumulation at the PM is toxic during heat stress. (**A**) heat-sensitivity analysis of wildtype, *Δart1*, *Δlyp1*, and *Δart1Δlyp1* mutant yeast cells. (**B**) Empty vector or plasmids encoding Lyp1-FLAG expressed from different promoters (pCPY, pLYP1, pADH1) were transformed into *Δart1Δlyp1* mutant yeast cells and the affect on heat tolerance was scored by growth at the indicated temperatures. Lyp1 expression level for each strain was analyzed by quantitative Western blot (Li-Cor). (**C**) Empty vector or plasmids encoding Lyp1-K10XR-FLAG (which lacks N-terminal lysine residues) expressed from different promoters (pCPY, pLYP1, pADH1, pTDH3) were transformed into wildtype yeast cells and the affect on heat tolerance was scored by growth at the indicated temperatures. (**D**) Flow cytometry was used to analyze PI staining of *Δart1Δlyp1* mutant yeast cells expressing Lyp1 from different promoters (pCPY, pLYP1, pADH1) following growth at 40°C for 3 hr. (**E**) Flow cytometry was used to analyze PI staining of wildtype yeast cells coordinately expressing Lyp1 from the TDH3 promoter and Art1 from different promoters (pART1, pCPY, pADH1, pTDH3) following growth at 40°C for 3 hr.

The following figure supplements are available for figure 5:

**Figure supplement 1**. Heat-sensitivity analysis of wildtype yeast cells expressing titrated levels of Lyp1.

*Figure 5. Continued on next page*

*Figure 5. Continued*

**Figure supplement 2**. Stability of wildtype Lyp1 or the K10XR mutant, which cannot be ubiquitinated on its N-terminal cytosolic tail.

**Figure supplement 3**. Fluorescence distribution of GFP-tagged wildtype Lyp1 (green, only in cells expressing Vph1-mCherry) or the K10XR mutant (green, only in cells without Vph1-mCherry) was analyzed following growth at 38°C for 2 hr.

(*Figure 5—figure supplement 1*). To further explore the toxicity of Lyp1 at high temperatures, we engineered a mutant of Lyp1 lacking cytosolic lysine residues (Lyp1-K10XR) that could be targets for Art1-Rsp5 ubiquitination. Importantly, the Lyp1-K10XR protein is efficiently delivered to the PM but cannot be internalized or degraded by the heat-induced endocytic response (*Figure 5—figure supplements 2 and 3*). High expression levels of Lyp1-K10XR resulted in increased temperature sensitivity even in wildtype cells (*Figure 5C*), demonstrating that accumulation of Lyp1 at the PM is toxic to the cell during heat stress. Importantly, the increased temperature sensitivity observed when Lyp1 accumulates at the PM correlates with increased PI staining during heat shock (*Figure 5D*). However, protection against this loss of PM integrity could be conferred upon increasing expression of Art1 (*Figure 5E*), which prevents integral membrane protein accumulation at the surface (*Lin et al., 2008*; *MacGurn et al., 2011*). Thus, our results indicate that accumulation of Lyp1 at the cell surface during heat stress is associated with loss of PM integrity and decreased cell viability. Under normal circumstances, heat stress triggers the recognition of Lyp1 by Art1, which targets its ubiquitination and removal from the PM, protecting the cell from toxic Lyp1 accumulation at the surface.

## The parallel architecture of protein quality control systems

Our results indicate that Rsp5 and the ART adaptor network function as part of a PM quality surveillance system, detecting integral membrane proteins at the cell surface as they become conformationally unstable and targeting their internalization and degradation. If the Rsp5-ART system is functioning as a bona fide quality control pathway at the PM, we would expect to observe genetic interactions with other quality control systems for integral membrane proteins along the secretory pathway (*Figure 6A*). To test this, we constructed a genetic array consisting of pairwise deletions known to affect protein quality control in the cytosol, nucleus, ER, Golgi complex, PM, endosome, and vacuole. We systematically scored each double deletion strain for PI staining following 3 hr of growth at 40°C to identify synthetic genetic interactions between quality control pathways (*Figure 6B*). In general, mutations that affected quality control of soluble cytosolic proteins did not interact genetically with mutations that affected quality control of integral membrane proteins. In contrast, we found that mutations in sequential quality control pathways for integral membrane proteins significantly enhanced the PM integrity defect observed for *Δart1* mutant cells following heat stress. For example, abrogation of either Golgi quality control (*Δvps10*, *Δgga1*, *Δtul1*) or ER quality control (*Δdoa10*, *Δhrd1*, *Δire1*) systems significantly enhanced the PM integrity defect observed for *Δart1* mutant cells during heat stress (*Figure 6C,D* and *Figure 6—figure supplement 1*). Furthermore, cells with defects in all three integral membrane quality control systems (ERQC, GQC, and PM quality control; *Δart1Δdoa10Δgga1* triple mutant cells) exhibited dramatic loss of PM integrity (~55% of total cell population) following 3 hr at 40°C (*Figure 6D*). Importantly, mutations that abrogate either ERQC or GQC, or abrogate both ERQC and GQC, do not result in temperature sensitive growth or significant PI-positive staining of cells following heat stress (*Figure 6D*), suggesting that PM quality control can compensate for defects in quality control along the secretory pathway. These genetic interactions underscore the critical protective function of sequential quality control systems along the secretory pathway and demonstrate how all three integral membrane quality control systems—ERQC, GQC, and PMQC—cooperate to protect the integrity of the PM during proteotoxic stress (*Figure 7*).

## Discussion

All cellular proteins are subject to turnover. The critical balance between synthesis and turnover is particularly important in very long-lived post-mitotic cells like neurons, where cellular proteostasis must be maintained over the entire lifetime of the organism. Thus, protein quality and quantity in all

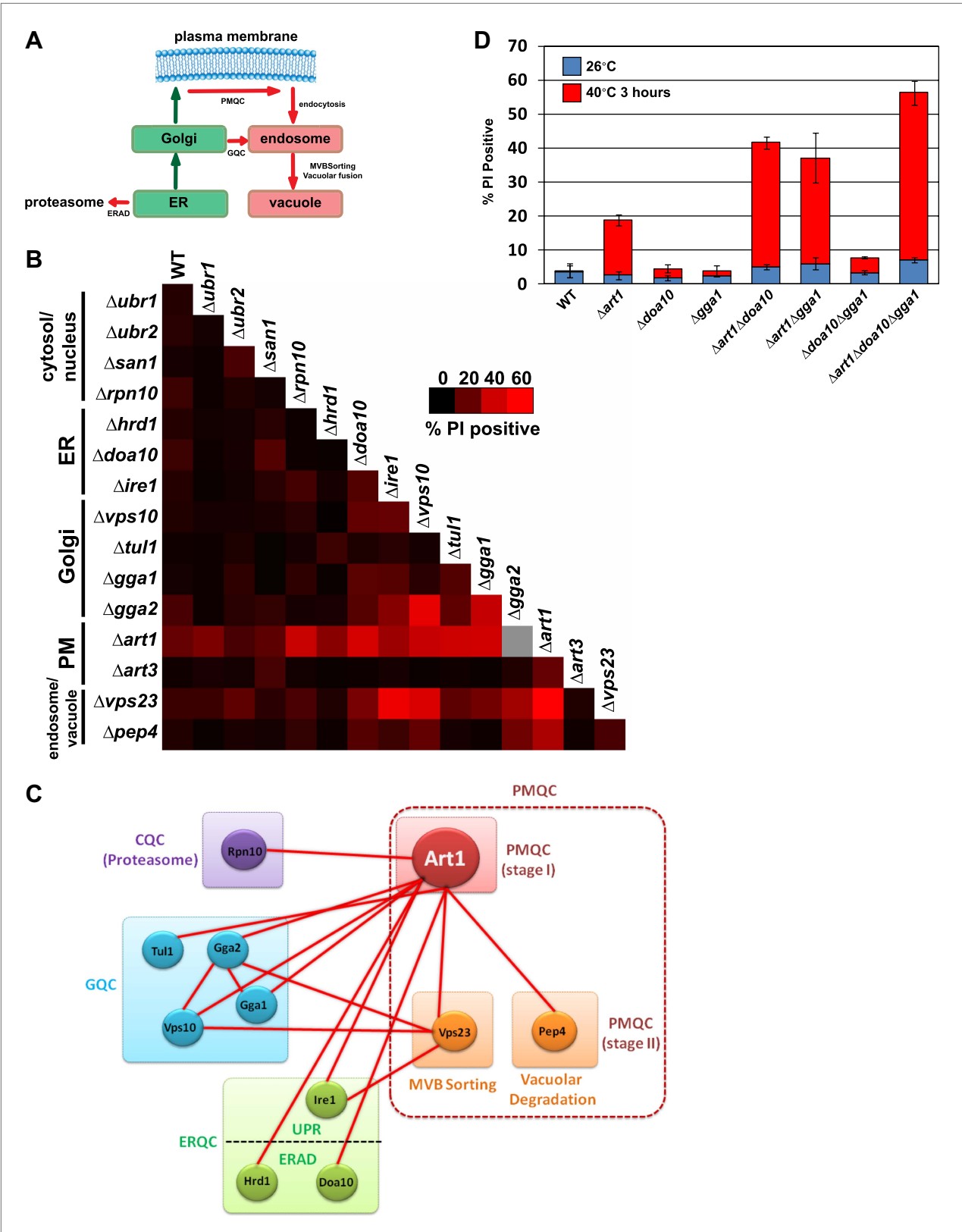

Figure 6. Systems level coordination of integral membrane protein quality control. (A) Model illustrating the major quality control mechanisms for integral membrane proteins: ERAD (proteasomal degradation), Golgi quality control (GQC; vacuolar/lysosomal degradation), and plasma membrane quality control (PMQC; vacuolar/lysosomal degradation). (B) Genetic array constructed for known quality control systems in the cell. Red color indicates
Figure 6. Continued on next page

*Figure 6. Continued*

the percentage of yeast cells staining PI positive following growth at 40°C for 3 hr (see key inset). Gray box indicates a strain which could not be obtained due to synthetic lethality in the SEY6210 background. (**C**) Node network illustrating all genetic interactions observed for protein quality control systems (cytosolic quality control, CQC; ER quality control, ERQC; Golgi quality control, GQC; PM quality control, PMQC) in *Figure 6B*. (**D**) Flow cytometry was used to analyze PI staining of the indicated strains following growth at 40°C for 3 hr.

The following figure supplements are available for figure 6:

**Figure supplement 1**. Heat-sensitivity analysis of wildtype and mutant yeast cells.

cellular compartments must be carefully managed both as a critical housekeeping function and to protect cellular integrity during proteotoxic stress. Protein quality control systems share several key features that define their function as mediators of cellular proteostasis. Such systems generally (i) have the ability to selectively recognize damaged or misfolded proteins, (ii) target a broad array of misfolded substrates for refolding or degradation, (iii) undergo induction or activation in response to proteotoxic stress, and (iv) protect cells from possible cell death in the presence of proteotoxic stresses. Based on these four criteria, we propose that the ART-Rsp5 adaptor network comprises a new protein quality control (PMQC) system that functions primarily in the recognition and targeted removal of misfolded integral membrane proteins at the PM.

## Recognition and removal of misfolded integral PM proteins during proteotoxic stress

An incredible yet poorly-understood property of protein quality control systems is the ability to preferentially recognize misfolded or aberrant proteins and target them either for re-folding or degradation. The potential number of disordered conformer states occupied by a protein as it misfolds is seemingly infinite, which poses a challenge for quality control systems to specifically recognize misfolded species with a broad range of substrates. In this study, we observed that many PM cargo proteins are turned over following heat stress, which is known to induce broad misfolding of proteins in the cell (*Fang et al., 2011*; *Theodoraki et al., 2012*). One thermolabile protein, the lysine transporter Lyp1, was shown to aggregate in isolated plasma membranes subject to heat stress, suggesting that heat stress triggers broad misfolding of cargo in the PM. We show that the ART-Rsp5 network (i) is required for the endocytic downregulation of cargo during proteotoxic stress and (ii) protects cells from the loss of PM integrity associated with accumulation of misfolded cargo at the surface. The recognition of misfolded proteins by ARTs is consistent with the finding that glycerol, a chemical chaperone that protects against protein misfolding, stabilized Lyp1 during heat stress without broadly affecting endocytosis (*Figure 1C* and *Figure 1—figure supplement 10*). Furthermore, we demonstrate that different cargo substrates targeted by the ART-Rsp5 network exhibit unique thermostability profiles and are recognized by specific ART-Rsp5 complexes, underscoring the cargo-intrinsic nature of substrate selection. Since each specific protein is expected to experience unique conformational dynamics in response to different proteotoxic stresses, these results are consistent with the targeting of proteins as they become conformationally disordered or misfolded (*Claessen et al., 2011*) and underscores the role of specific ART proteins in recognizing distinct misfolded signatures.

Although it is still poorly understood how quality control machinery discriminates between folded and unfolded clients, recent studies have established two general mechanisms for targeting misfolded proteins. One mechanism involves targeting of misfolded proteins by binding of E3 ubiquitin ligases to molecular chaperones. For example, the E3 ubiquitin ligase CHIP binds to Hsp70, which effectively targets chronically misfolded proteins for ubiquitination and subsequent proteasomal degradation (*Meacham et al., 2001*; *Okiyoneda et al., 2010*). While it is possible that the ART-Rsp5 is targeted to misfolded integral membrane proteins via chaperones, candidate chaperones for the detection of misfolded integral membrane proteins are not obvious. A second quality control strategy involves the recognition of misfolded proteins directly by E3 ubiquitin ligases via intrinsically disordered regions with the potential to target exposed hydrophobic domains (*Fredrickson et al., 2011*). For example, the nuclear E3 ubiquitin ligase San1 is capable of targeting the degradation of a broad range of soluble nuclear proteins by a mechanism that requires intrinsically disordered segments of the San1 protein (*Rosenbaum et al., 2011*). Indeed, while San1 is predicted to contain 12 disordered regions, the complete network of ART adaptors is predicted to contain 173 disordered regions (some greater than

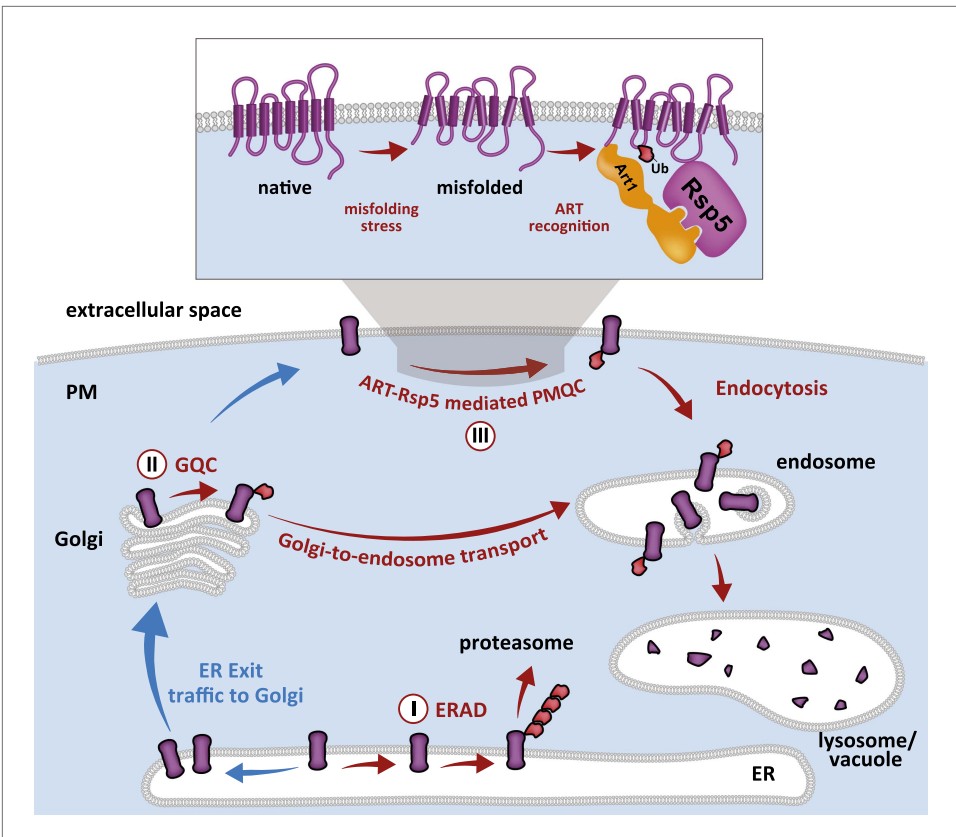

**Figure 7**. Cellular mechanisms of integral membrane protein quality control. Integral membrane proteins are subject to sequential quality control mechanisms including ERAD in the ER (I), GQC in the Golgi (II), and ART-Rsp5 mediated PMQC at the PM (III).

100 amino acids in length) demonstrating the potential capacity for this system to directly detect misfolded proteins. Ultimately, recognition of integral membrane protein misfolding may require mechanisms distinct from those employed in the recognition of misfolded soluble proteins. For example, it is possible that misfolding stress may trigger the aggregation of membrane spanning domains and the ART-Rsp5 network may recognize biochemical features of this type of aggregation. Future studies will need to address structural mechanisms for the recognition of misfolded PM proteins by the ART-Rsp5 network.

## Protecting cellular integrity by endocytic clearance of integral PM proteins

Ultimately, the most critical feature of protein quality control systems is that they are required to protect cells during proteotoxic stress and prevent possible cell death. We found in the course of our investigation that abrogation of the ART-Rsp5 network results in hypersensitivity to heat stress that is associated with accumulation of proteins at the cell surface and loss of PM integrity. Thus, the ART-Rsp5 network plays a critical role in protecting the cell—and in particular, protecting the plasma membrane—during conditions of proteotoxic stress. It is unclear exactly how accumulation of misfolded proteins at the cell surface could result in loss of PM integrity, but we speculate that nutrient transporters and ion channels—whose primary function is to permit selective transport of small molecules through the PM—are a particular liability given their potential to allow non-specific leakage if not properly folded. Indeed, the function of many PM channels and transporters requires transition between multiple conformational states in order to facilitate transfer of molecules between the extracellular space and the cytosol. These conformational transitions must be carefully controlled to prevent loss of critical ion gradients, especially during conditions of proteotoxic stress where excessive conformational flux may present a significant liability to PM integrity. Alternatively, the accumulation of misfolded integral membrane proteins could result in aggregation which broadly disrupts the function of proteins at the surface. Understanding not only the mechanism for recognition and removal of misfolded integral PM

proteins but also the basis of their toxicity will contribute to our overall understanding of proteostasis networks and help shed light on the cytotoxicity associated with protein misfolding disorders.

In many cases, protein quality control systems are induced or activated during conditions of proteotoxic stress. For example, the heat shock response activates a transcriptional program that increases expression of chaperones, ubiquitination machinery, and other proteins dedicated to dealing with protein folding stress. Similarly, the unfolded protein response is induced following ER stress, resulting in the induction of ER chaperones and ERAD machinery (*Walter and Ron, 2011*). While the ART-Rsp5 network does not appear to be induced transcriptionally, we find that it does appear to be activated in response to heat stress. Specifically, we find that heat stress triggers PM translocation of Art1 (*Figure 4—figure supplement 2*). Our previous studies revealed that TORC1 signaling induced Art1 dephosphorylation which triggered PM translocation, however we find no evidence that heat stress affects the phosphorylation status of Art1 (unpublished results). Thus, the mechanistic basis of Art1 translocation to the PM in response to heat stress remains to be elucidated.

Although removal of misfolded integral PM proteins is critical to protect PM integrity during proteotoxic stress, we speculate that similar quality control mechanisms may also serve important housekeeping functions during normal cell growth. Indeed, all integral PM proteins have a functional lifetime and must ultimately be removed and degraded. Even in the absence of proteotoxic stress, accumulation of these proteins could be detrimental to the cell. Indeed, many human channelopathies, which include many neurological or cardiac disorders, are very likely caused by mutations that affect both the quality and quantity of specific ion channels at the cell surface (*Kullmann, 2010*; *Rougier et al., 2010*).

## The parallel circuitry of membrane protein quality control systems

Our results indicate that quality control systems for integral membrane proteins—including ERQC, GQC, and PMQC—cooperate to limit the accumulation of misfolded proteins at the PM. Although ERAD is the best understood quality control system that targets integral membrane proteins for degradation, we found that defects in ERAD did not significantly affect thermotolerance in the presence of Art1 but did exacerbate the loss of PM integrity in the absence of Art1. GQC, while still poorly understood, has also recently been shown to protect the cell from proteotoxic stress by capturing misfolded proteins at the Golgi and diverting them to the endosomal system for degradation in the lysosome (*Wang et al., 2011*). We found that defects in all three sequential pathways—ERQC, GQC, and PMQC—resulted in severe PM integrity defects when subject to misfolding stress, suggesting they function as spatially distinct yet mechanistically overlapping protein quality control systems. We cannot exclude the possibility that the synthetic genetic interactions observed between the ART-Rsp5 network, GQC, and PMQC may result not from defects in sequential quality control systems but instead from nonspecific effects of impairing quality control at multiple organelles in the cell. Nevertheless, we propose that the ART-Rsp5 network in yeast functions as the major pathway for clearance of PM proteins that exceed their functional lifetime or misfolded proteins that traffic to the surface by escaping ER and Golgi QC. Understanding the coordination of compartment-specific quality control systems along the secretory and endocytic pathways has the potential to contribute to improved therapeutic strategies for protein misfolding diseases like CFTR and other channelopathies (*Kullmann, 2010*; *Rougier et al., 2010*).

## Plasma membrane quality control (PMQC) systems and protection from proteotoxic stress

The earliest studies reporting the existence of post-ER quality control of integral membrane proteins provided evidence that misfolding of mutant membrane proteins in yeast is triggered by either high temperature (*Jenness et al., 1997*; *Li et al., 1999*; *Liu et al., 2006*) or changes in PM lipid composition (*Wang and Chang, 2002*; *Lauwers et al., 2007*). Recent studies have provided evidence of a PM quality control system in mammalian cells that contributes to the ubiquitination and lysosomal trafficking of misfolded ΔF506 CFTR (*Okiyoneda et al., 2010*) as well as thermolabile synthetic cargoes (*Apaja et al., 2010*). Both studies converged on a common mechanism that involves recognition of misfolded soluble cytosolic domains by chaperones (Hsp70/Hsc70/Hsp90) which recruit CHIP, an E3 ubiquitin ligase. While the yeast genome does not encode a homolog of mammalian CHIP, the analogous proteins, Ubr1 and Ubr2, are also required for the degradation of misfolded proteins (*Khosrow-Khavar et al., 2012*). However, Ubr1 and Ubr2 are dispensable for the heat-induced endocytic response associated with the turnover of integral PM proteins (*Figure 6B* and data not shown). Instead, our results indicate that the recognition of misfolded integral membrane proteins in yeast is mediated by ART adaptor proteins. It

is tempting to speculate that a similar peripheral quality control function may be mediated by the ARRDC family of proteins in mammalian cells, which encode proteins similar to ARTs with an N-terminal arrestin-domain and a C-terminal domain containing multiple PY motifs capable of binding to members of the Nedd4 family of ubiquitin ligases (*Patwari and Lee, 2012*). In the context of a multicellular organism, these membrane quality control systems may prevent the affects of accumulated protein damage over time and thus ultimately protect cells from premature aging and cell death.

## Materials and methods

### Fluorescence microscopy

Yeast cells expressing fluorescent fusion proteins were grown to mid-log phase in synthetic media. Microscopy was performed using a fluorescence microscope (DeltaVison RT; Applied Precision, Issaquah, WA) equipped with FITC and rhodamine filters. Images were captured with a digital camera (Cool Snap HQ; Photometrics, Tucson, AZ) and deconvolved using softWoRx 3.5.0 software (Applied Precision, Issaquah, WA).

### Analysis of cellular protein expression levels

Yeast cells expressing epitope-tagged protein were grown to mid-log phase in synthetic media. 5 $OD_{600}$ equivalents of mid-log cells pretreated at the indicated temperatures were harvested by precipitation in 10% trichloroacetic acid (TCA). Precipitates were washed in acetone, aspirated, resuspended in lysis buffer (150 mM NaCl, 50 mM Tris pH7.5, 1 mM EDTA, 1% SDS), and mechanically lysed with glass beads. Protein sample buffer (150 mM Tris pH 6.8, 6M Urea, 6% SDS, 10% beta-mercaptoethanol, 20% Glycerol) was added and extracts were analyzed by SDS-PAGE and immunoblotting with anti-FLAG (Sigma, St. Louis, MO) antibody.

### Detergent solubility analysis and sucrose gradients

Yeast cells were disrupted by bead-beating in lysis buffer (PBS + 1 mM EDTA + protease inhibitors). Lysates were centrifuged at 500×*g* for 5 min to remove unbroken cells and nuclei. Cleared lysates were then spun at 13,000×*g* for 15 min to isolate P13 fractions. P13 fractions were washed once, resuspended in lysis buffer and incubated for 30 min at either 26°C or 40°C then placed on ice. Detergent (n-dodecyl-β-D-maltopyranoside; Affymetrix, Santa Clara, CA) was added to 1% and samples were incubated for 30 min at 4°C. Samples were then spun at 40,000×*g* for 30 min to isolate S40 and P40 fractions. To assay protein aggregation, S40 and P40 fractions were loaded onto 20%/40% sucrose step gradients and spun at 55,000×*g* for 60 min. Top and bottom fractions of the sucrose gradient were isolated, precipitated in 10% TCA and analyzed by Western blot.

### Plasma membrane integrity assay

Yeast strains were grown to early-log phase (around 0.2 $OD_{600}$) in synthetic media, shifted to 40°C and cultured for an additional 3 hr. 1 $OD_{600}$ equivalent of cells was pelleted and resuspended in PBST (0.01% Tween 20) and cells were stained with propidium iodide (Sigma) for 20 min. Cells were then washed twice with ddH$_2$0 and analyzed by flow cytometry.

## Acknowledgements

We thank A Chang, S Moye-Rowley, and L Miller for reagents and helpful advice. We also thank M Chin for technical assistance. We are grateful to S Qian and C Fromme for critical reading of the manuscript. We also thank members of the Emr lab for helpful discussions, especially PC Hsu.

## Additional information

### Funding

| Funder | Grant reference number | Author |
|---|---|---|
| NIH K99 Pathway to Independence Award | 1K99GM101077 | Jason A MacGurn |
| Cornell University Research Grant | | Scott Emr |

The funders had no role in study design, data collection and interpretation, or the decision to submit the work for publication.

## Author contributions

YZ, Conception and design, Acquisition of data, Analysis and interpretation of data; JAM, Conception and design, Acquisition of data, Analysis and interpretation of data, Drafting or revising the article; ML, Generated strains and reagents for the genetic array, Carried out the genetic array experiments, Analysis and interpretation of data; SE, Conception and design, Analysis and interpretation of data, Drafting or revising the article

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
