## [Decision Letter]

Thank you for choosing to send your work entitled “The ART-Rsp5 ubiquitin ligase network comprises a plasma membrane quality control system that protects cells from proteotoxic stress” for consideration at *eLife*. Your article has been favorably evaluated by a Senior editor and 3 reviewers, one of whom is a member of our Board of Reviewing Editors.

The following individuals responsible for the peer review of your submission want to reveal their identity: Raymond Deshaies (Reviewing editor); Peter Walter (peer reviewer). The Reviewing editor and the other reviewers discussed their comments before we reached this decision, and the Reviewing editor has assembled the following comments to help you prepare a revised submission.

The manuscript by Zhao et al. is an exciting study that describes the existence of a general mechanism for the detection and clearance of misfolded proteins at the plasma membrane of yeast cells. The existence of such a mechanism in animal cells has been documented in papers from the Lukacs' laboratory. However, the present work from the Emr group extends these findings in a very substantial manner by showing how the ubiquitin-driven endocytosis machinery (involving the Art protein family members and the ubiquitin ligase Rsp5) surveys the plasma membrane for defective proteins and removes them. In addition, the authors document the consequences that befall cells when this mechanism fails.

The reviewers are in unanimous agreement that this work is well suited for publication in *eLife*. However, two separate concerns were raised that should be addressed prior to publication.

1) Based on the data in Figure 6 it is argued that ER, Golgi, and PM quality-control pathways act in parallel to ensure the quality of cell surface proteins. This conclusion is based on synergistic effects of combining mutations in two of the three, or all three, pathways. However, there are two issues with these data. First, the *Δart1* that is the backbone of the analysis in Figure 6B shows fairly extensive genetic interaction with multiple other genes, including *Δpep4*, *Δrpn10*, and *Δvps23*. The relative lack of specificity somewhat undermines the conclusions. Second, the quantitative analysis of genetic interactions in panel D is not compelling in its current form. Given the lack of statistical analysis, it is not possible to tell whether the relatively small synergy reported (compared to the additive effects expected for completely non-interacting genes) is significant. A further problem with this experiment is that results are reported only for cells shifted to 40 °C for 3 hr. Thus it is not possible to know how much of the signal seen here represents a higher proportion of dead cells during growth in culture at low temperature vs increased permeabilization upon heat stress. This section of the paper either needs to be strengthened or the figure should be removed.

2) A critical aspect of the study is the conclusion that Rsp5-Arts act at the plasma membrane by stimulating endocytosis in response to heat stress. However, in some cases (e.g., Figure 1, Can1-GFP and Mup1-GFP), although the vacuolar GFP signal is clear in the heat-treated samples, there is not an obvious corresponding decrease in plasma membrane signal that might be expected for endocytic down-regulation. This raises the possibility that heat stress is primarily acting to trigger sorting of newly synthesized plasma membrane proteins from the TGN, without them ever having reached the plasma membrane. This possibility should be ruled out by appropriate control experiment(s). One possibility would be to evaluate the effect of heat stress on localization of GFP chimeras in *Δend3* or *Δrvs167*.

---

## [Author Response]

*1) Based on the data in Figure 6 it is argued that ER, Golgi, and PM quality-control pathways act in parallel to ensure the quality of cell surface proteins. This conclusion is based on synergistic effects of combining mutations in two of the three, or all three, pathways. However, there are two issues with these data. First, the* Δart1 *that is the backbone of the analysis in Figure 6B shows fairly extensive genetic interaction with multiple other genes, including* Δpep4*,* Δrpn10*, and* Δvps23. *The relative lack of specificity somewhat undermines the conclusions. Second, the quantitative analysis of genetic interactions in panel D is not compelling in its current form. Given the lack of statistical analysis, it is not possible to tell whether the relatively small synergy reported (compared to the additive effects expected for completely non-interacting genes) is significant. A further problem with this experiment is that results are reported only for cells shifted to 40 °C for 3 hr. Thus it is not possible to know how much of the signal seen here represents a higher proportion of dead cells during growth in culture at low temperature versus increased permeabilization upon heat stress. This section of the paper either needs to be strengthened or the figure should be removed*.

Although the data presented in Figure 6 suggest that the three major quality control systems for integral membrane proteins–ERQC, GCQ, and PMQC–are acting sequentially to limit accumulation of misfolded proteins at the surface, we agree with the reviewers that the data could also be consistent with alternative interpretations and conclusions. Although we cannot explain all the results presented in the genetic array (Figure 6B) we believe that these results (including genetic interactions of *Δart1* with *Δpep4, Δrpn10*, and *Δvps23)* are of general interest to both the protein quality control and membrane trafficking fields. Thus, we believe that the best way to address the reviewers' concern is to leave the data in the manuscript but to acknowledge the possibility for alternative interpretations wholly consistent with the data. Additionally, we have added a node network diagram that illustrates the synthetic genetic interactions observed between different nodes in the quality control array. We believe this node network diagram illustrates the inter-connectedness of the different sequential quality control systems.

We also agree with the reviewers that some of the data presented in Figure 6D would benefit from more rigorous analysis. To this end, we have now included additional analysis of PI staining at 26°C, which illustrates that background PI staining of these strains at ambient temperature is uniformly low and that the high PI staining observed following growth at 40°C does not result from dead cells pre-existing in the culture prior to temperature shift. This analysis also highlights the synergistic effects of abrogating multiple membrane protein quality control pathways: loss of Art1 confers a mild heat-induced cell permeability phenotype (19% of cells), but when combined with mutations that abrogate ERAD and GQC this phenotype is enhanced three-fold (57% of cells). We have added similar analysis for other PI staining experiments, including Figures 3C and 5D.

*2) A critical aspect of the study is the conclusion that Rsp5-Arts act at the plasma membrane by stimulating endocytosis in response to heat stress. However, in some cases (e.g., Figure 1, Can1-GFP and Mup1-GFP), although the vacuolar GFP signal is clear in the heat-treated samples, there is not an obvious corresponding decrease in plasma membrane signal that might be expected for endocytic down-regulation. This raises the possibility that heat stress is primarily acting to trigger sorting of newly synthesized plasma membrane proteins from the TGN, without them ever having reached the plasma membrane. This possibility should be ruled out by appropriate control experiment(s). One possibility would be to evaluate the effect of heat stress on localization of GFP chimeras in Δend3 or Δrvs167*.

We agree with the reviewers that it is important to establish that the ART-Rsp5 network acts at the plasma membrane in response to heat stress. We also agree with the reviewers that some portion of the vacuolar signal during heat stress could result from biosynthetic delivery to the vacuole without ever having reached the plasma membrane (i.e., though Golgi quality control or GQC). Indeed, we believe that both PM and Golgi quality control pathways operate during proteotoxic stress to deliver cargo to the vacuole for degradation and thereby limit surface protein accumulation. However, the rapid turnover observed for many PM proteins following exposure to heat stress (Figure 1B) seems wholly consistent with endocytic downregulation and could not be explained by the turnover of newly synthesized cargo via Golgi-to-endosome trafficking.

In order to demonstrate endocytic downregulation (i.e., that the increased vacuolar signal observed during heat stress corresponds to a decrease in signal from the plasma membrane), we quantified heat-induced changes in cargo abundance at the cell surface in two ways. First, for both Mup1-GFP and Can1-GFP we quantified PM fluorescence intensity (per unit area) at 26°C and 40°C over many cells (n ≥ 30) and found a significant decrease in PM fluorescence intensity following exposure to heat stress (Figure1–figure supplement 1), which corresponds to increased vacuolar fluorescence intensity. Second, to more accurately quantify surface abundance over a population of yeast cells, we used Mup1-pHluorin, which does not fluoresce in the acidic environment of the vacuole. Thus, for cells expressing Mup1-pHluorin, almost all of the fluorescence signal originates from the PM (Figure 1–figure supplement 2, top panel). Flow cytometry analysis revealed an approximately 50% decrease in surface fluorescence following exposure of the culture to heat stress, which is consistent with the decrease observed for Mup1-GFP (Figure 1–figure supplement 1) and Mup1-FLAG (Figure 1–figure supplement 3). Thus, we believe that these experiments indicate loss of cargo from the PM consistent with endocytic downregulation.